# Political factors and arrangements influencing primary health care financing and resource allocation: A scoping review protocol

**Henrique Sant'Anna Dias**[1]☯*, **Adelyne Maria Mendes Pereira**[2‡], **Elisabete de Fátima Polo de Almeida Nunes**[3☯], **Caroline Pagani Martins**[3‡], **Marcela Castilho**[3‡], **Fernanda de Freitas Mendonça**[3‡], **Luciana Dias de Lima**[2‡]

**1** Research and Innovation Vice-Direction, Sergio Arouca National School of Public Health, Rio de Janeiro, RJ, Brazil, **2** Department of Health Planning and Administration, Sergio Arouca National School of Public Health, Rio de Janeiro, Rio de Janeiro, Brazil, **3** Department of Collective Health, Londrina State University, Londrina, Paraná, Brazil

☯ These authors contributed equally to this work.
‡ The authors also contributed equally to this work.
* henriquesesrj@gmail.com

**Data Availability Statement:** No datasets were generated or analysed during the current study

## Abstract

### Introduction

Primary health care is a key element in the structuring and coordination of health systems, contributing to overall coverage and performance. PHC financing is therefore central in this context, with variations in sufficiency and regularity depending on the "political dimension" of health systems. Research that systematically examines the political factors and arrangements influencing PHC financing is justified from a global and multidisciplinary perspective. The scoping review proposed here aims to systematically map the evidence on this topic in the current literature, identifying groups, institutions, priorities and gaps in the research.

### Methods and analysis

A scoping review will be conducted following the method proposed by Arksey and O'Malley to answer the following question: What is known from the literature about political factors and arrangements and their influence on and repercussions for primary health care financing and resource allocation models? The review will include peer-reviewed papers in Portuguese, English or Spanish published between 1978 and 2023. Searches will be performed of the following databases: Medline (PubMed), Embase, BVS Salud, Web of Science, Scopus and Science Direct. The review will follow the Preferred Reporting Items for Systematic Reviews and Meta-Analyses extension for Scoping Reviews checklist. Inclusion and exclusion criteria will be used for literature screening and mapping. Screening and data charting will be conducted by a team of four reviewers.

### Registration

This protocol is registered on the Open Science Framework (OSF) platform, available at https://doi.org/10.17605/OSF.IO/Q9W3P

protocol. All data, documents and reports generated from implementation of proposed scoping review will be available from Fiocruz Repository (www.arca.fiocruz.br; https://arcadados.fiocruz.br/).

**Funding:** This work was supported by the FIOCRUZ Program for Strengthening Primary Health Care in Brazil – PMA 2019 (Fiocruz/VPPCb; grant number 25380.101539/2019- 05) The funders had no role in study design, data collection and analysis, decision to publish, or preparation of the manuscript.

**Competing interests:** The authors have declared that no competing interests exist.

## 1. Introduction

Primary care is the point of first contact with the health system and aims to bring comprehensive health care as close as possible to where people live [1,2]. Analyses addressing the attributes of PHC in different contexts, countries and health systems are important as they reveal variations in care models, management and organization [3,4].

The literature highlights the importance of primary care for the coordination of care and points of care (actions and services) within the health system, contributing to overall performance against health indicators [5–8]. Health outcomes bear a relation with the predominant 'type of PHC', with good outcomes being positively associated with comprehensive approaches as opposed to selective PHC [9–11]. Robust, comprehensive PHC with a high level of coverage and that promotes effective coordination between primary care and other levels of care therefore produces better outcomes.

Since the Alma-Ata Declaration, endorsed at the International Conference on Primary Health Care in 1978 [12], PHC has been affirmed as the cornerstone of public health systems, gaining prominence in health policymaking [13–17]. Discussions surrounding the performance and effectiveness of PHC include aspects such as public financing and the adoption of care models in the midst of health reform processes and the expansion of coverage of primary care [18–22].

Proposals aimed at ensuring regular adequate funding can be found in initiatives developed by multilateral organizations such as the WHO [23] and PAHO [24,25] and scientific bodies, including The Lancet Global Health Commission on financing primary health care [26]. Revamped proposals were also one of the outcomes of the Astana Conference (2018), which upheld the position of the Alma-Ata Declaration in relation to the importance of PHC for the expansion of the coverage of health actions and services [27].

Studies on health financing involve aspects related to the composition of revenue and health spending, including resource allocation mechanisms and forms of payment for health actions and services [28–32]. The definition of financing models is central to promoting financial protection, sustainability and improvements in health system outcomes in countries with differing levels of income located across different regions. This process is mediated by different actors, sectors and civil society, market and state organizations, with variations in the scale and balance of public-private partnerships in the sector [33–35].

Health financing results from economic, financial, social and political contingencies and challenges, acquiring different designs, formats and perspectives depending on the political and government system in place, intergovernmental relations and relations between the branches of the federal government. Economic rules that determine fiscal policy and budgeting are also relevant, especially when it comes to identifying sources of funding and financial resource allocation priorities. A health system's organizational and institutional design, its actors, bodies, services, policies and guiding principles also constitute an important set of variables that confer materiality to financing when considering decisions regarding the earmarking resources and health expenditures.

PHC policies, and consequently the financing of these policies, are embedded in the political and institutional dynamics that shape different health systems around the world. Political factors and arrangements connected to PHC express the degree of priority given to this level of care on political and government agendas, which in turn depends on the government and legislative environment in which the courses of action and policy choices that characterize health systems and actions, and their multiple dimensions, scope, objectives and reach, are determined.

The aim of the proposed review is to map the literature on the political factors and arrangements that influence primary health care financing and resource allocation.

**Patient and public involvement**

Patients or the public were not involved in the design, or conduct, or reporting, or dissemination plans of our scoping review protocol

## 2. Methods and analyses

A scoping review will be conducted to map the existing literature on the political factors and arrangements that influence primary health care financing and resource allocation. The study will be developed using the methodological framework proposed by Arksey and O'Malley [36] and recommendations subsequently developed by Levac *et al* [37]. This protocol was developed in accordance with the guidance published by the Joanna Briggs Institute [38] and the PRISMA-ScR checklist [39]. The proposed study does not require ethical approval.

### a) Identifying the research question

The following research question was formulated: what is known from the literature about political factors and arrangements and their influence on and repercussions for primary health care financing and resource allocation models?

As proposed by the JBI, the research question was formulated using the PCC mnemonic, which stands for Population, Concept, and Context. The population comprises political and government authorities and managers and officials working in health systems and related organizations, as well as other actors involved in health policy-making.

The key concept refers to the political factors that influence financing and resource allocation models in different health system configurations. Studies that will therefore be considered for inclusion in the review include essentially inter and multidisciplinary political analyses documenting a broad range of models, approaches and theories.

Relevant studies include those highlighting political processes, political and social institutions, contexts, actors, rules, disputes, interests, and political arenas and systems. The following types of studies will also be considered for inclusion: studies discussing political and (inter) governmental relations, power relations and fiscal federalism; policy analyses and studies investigating policy-making and implementation; studies adopting the political determinants of health approach [40–42]; studies focusing on political science and health [43,44]; studies adopting institutionalist approaches [45–47]; and studies involving the political economy of health [48,49].

The following political factors will be focused on: (1) at the governmental level, the composition of the government, its political and ideological orientations and priority health agendas; and (2) at the legislative level, the process related to economic and health regulation, involving the presence of lobbies, defence coalitions, legislative and party activities and representations of interests of different social groups.

The influence of politics (here defined as "political factors and arrangements") on models of health financing encompasses the definition, composition and type of sources of funding and strategies, and mechanisms employed to achieve model sufficiency and sustainability associated with health system configurations and population health status. When it comes to resource allocation models, these factors are expressed in health spending decisions (which costs and operational aspects of spending), efficiency, planning, management, control and quality. The context is primary health care (PHC), which plays a central role in public health system structuring and reform oriented towards expanding access and coverage and ensuring the universal right to health.

The population, concept and context that will guide the scoping review method presented in this protocol are outlined in Table 1.

**Table 1. Population, concept and context for the proposed scoping review Political factors and arrangements influencing primary health care financing and resource allocation.**

| | |
|---|---|
| **Population** | Political and government authorities, managers and officials working in health systems and related organizations in different countries |
| **Concept** | Political factors influencing health financing and resource allocation models, such as political and ideological orientations, agendas, composition and government health definitions; and legislative process of economic and health regulation, lobbies, defence coalitions, parliamentary and party activities and representation of interests of different social groups. |
| **Context** | Primary health care (PHC) |

Source: The authors.

## b) Identification of relevant studies

The broad range of disciplines and terminology covered by the key concept means that it will be necessary to use a diverse range of databases and search terms. We developed a process of continuous refinement of terms and keywords identified in preliminary searches using DeCS/MeSH descriptors to assess the adequacy of the terms, descriptors and keywords that will be used in the search strategies.

Table 2 presents the content resulting from this process grouped according to study concept and context. The box includes the Boolean operators 'AND' and 'OR' used to create the search strategies, which were developed with the support of a qualified librarian and tailored to the specific characteristics of each database.

As recommended by the JBI, a three-step search strategy will be used, beginning with a search of the databases Medline (PubMed) and Embase, followed by a second search using all identified keywords and terms undertaken across the remaining selected databases: BVS Salud, Web of Science, Scopus and Science Direct. The searches will then be followed by an analysis of the terms and keywords contained first in the title and then in the abstract of the retrieved papers, and of the index terms used to describe the articles. Finally, the citations and reference lists of the identified publications will be searched for additional sources, ensuring the inclusion of a wide range of studies for review.

**Table 2. Preliminary descriptors, keywords and search terms according to study concept and context, and the Boolean operators of the suggested standardized search strategy.**

| Concept/context | Descriptors/keywords/terms |
|---|---|
| **Politics, political and institutional arrangements and political factors** | ("political and institutional arrangement" OR "political arrangement" OR "institutional arrangement" OR "political-institutional arrangement" OR "politics" OR "political factor" OR "political system" OR "public policy" OR "governmental organization") |
| AND | |
| **Health financing and resource allocation** | ("financing" OR "resources allocation" OR "spending" OR "expenditure" OR "funding" OR "allocative efficiency" OR "allocation of health resources" OR "national health accounts" OR "health system financing" OR "capitation fee" OR "pay-for-performance" OR "payment system" OR "fee-for-service plans" OR "capitation" OR "reimbursement" OR "incentive") |
| AND | |
| **Primary health care** | ("primary care" OR "primary healthcare" OR "primary health care" OR "primary health" OR "PHC") |

Source: The authors.

The following search filters will be used: language (English, Spanish and Portuguese); and time period (1978 to 2023). The year 1978 is considered a milestone for public health due to the Alma-Ata Declaration, which emphasizes the central role played by PHC in the re(organi-zation) of health systems, policies and care [50].

The proposed search strategies for each selected database are detailed in Table 3.

**Table 3. Proposed search strategies by selected database.**

| Database | Search |
|---|---|
| Medline (PubMed) | (("political and institutional arrangement" OR "political arrangement" OR "institutional arrangement" OR "political-institutional arrangement" OR "politics" OR "political factor" OR "political system" OR "public policy" OR "governmental organization") AND ("financing" OR "resources allocation" OR "spending" OR "expenditure" OR " funding" OR "allocative efficiency" OR "allocation of health resources" OR "national health accounts" OR "health system financing" OR "capitation Fee" OR "pay-for-performance" OR "payment system" OR "fee-for-service plans" OR "capitation" OR "reimbursement" OR "incentive")) AND ("primary care" OR "primary healthcare" OR "primary health care" OR "primary health" OR "PHC") Filters: English, Portuguese, Spanish, MEDLINE, from 1978–2023 Sort by: Publication Date (((("political"[All Fields] OR "politically"[All Fields] OR "politics"[MeSH Terms] OR "politics"[All Fields] OR "politic"[All Fields]) AND ("academies and institutes"[MeSH Terms] OR ("academies"[All Fields] AND "institutes"[All Fields]) OR "academies and institutes"[All Fields] OR "institute"[All Fields] OR "institutes"[All Fields] OR "health facilities"[MeSH Terms] OR ("health"[All Fields] AND "facilities"[All Fields]) OR "health facilities"[All Fields] OR "institution"[All Fields] OR "institute"[All Fields] OR "institute s"[All Fields] OR "instituted"[All Fields] OR "instituting"[All Fields] OR "institution s"[All Fields] OR "institutional"[All Fields] OR "institutions"[All Fields] OR "institutes"[All Fields]) AND ("arrange"[All Fields] OR "arranged"[All Fields] OR "arrangement"[All Fields] OR "arrangements"[All Fields] OR "arranges"[All Fields] OR "arranging"[All Fields])) OR (("political"[All Fields] OR "politically"[All Fields] OR "politics"[MeSH Terms] OR "politics"[All Fields] OR "politic"[All Fields]) AND ("arrange"[All Fields] OR "arranged"[All Fields] OR "arrangement"[All Fields] OR "arrangements"[All Fields] OR "arranges"[All Fields] OR "arranging"[All Fields])) OR "institutional arrangement"[All Fields] OR ("political institutional"[All Fields] AND ("arrange"[All Fields] OR "arranged"[All Fields] OR "arrangement"[All Fields] OR "arrangements"[All Fields] OR "arranges"[All Fields] OR "arranging"[All Fields])) OR "politics"[All Fields] OR "political factor"[All Fields] OR "political system"[All Fields] OR "public policy"[All Fields] OR "governmental organization"[All Fields]) AND ("financing"[All Fields] OR "resources allocation"[All Fields] OR "spending"[All Fields] OR "expenditure"[All Fields] OR "funding"[All Fields] OR "allocative efficiency"[All Fields] OR "allocation of health resources"[All Fields] OR "national health accounts"[All Fields] OR "health system financing"[All Fields] OR "capitation Fee"[All Fields] OR "pay-for-performance"[All Fields] OR "payment system"[All Fields] OR "fee-for-service plans"[All Fields] OR "capitation"[All Fields] OR "reimbursement"[All Fields] OR "incentive"[All Fields]) AND ("primary care"[All Fields] OR "primary healthcare"[All Fields] OR "primary health care"[All Fields] OR "primary health"[All Fields] OR "PHC"[All Fields])) AND ((Medline[Filter]) AND (1978:2023[pdat]) AND (English[Filter] OR Portuguese[Filter] OR Spanish[Filter])) |
| Embase | ('political and institutional arrangement' OR 'political arrangement' OR 'institutional arrangement' OR 'political-institutional arrangement' OR 'politics' OR 'political factor' OR 'political system' OR 'public policy' OR 'governmental organization') AND ('financing' OR 'resources allocation' OR 'spending' OR 'expenditure' OR 'funding' OR 'allocative efficiency' OR 'allocation of health resources' OR 'national health accounts' OR 'health system financing' OR 'capitation fee' OR 'pay-for-performance' OR 'payment system' OR 'fee-for-service plans' OR 'capitation' OR 'reimbursement' OR 'incentive') AND ('primary care' OR 'primary healthcare' OR 'primary health care' OR 'primary health' OR 'phc') AND ([english]/lim OR [portuguese]/lim OR [spanish]/lim) AND [1978–2023]/py |
| BVS Salud | ("political AND institutional arrangement" OR "political arrangement" OR "institutional arrangement" OR "political-institutional arrangement" OR "politics" OR "political factor" OR "political system" OR "public policy" OR "governmental organization") AND ("financing" OR "resources allocation" OR "spending" OR "expenditure" OR "funding" OR "allocative efficiency" OR "allocation of health resources" OR "national health accounts" OR "health system financing" OR "capitation fee" OR "pay-for-performance" OR "payment system" OR "fee-for- service plans" OR "capitation" OR "reimbursement" OR "incentive") AND ("primary healthcare" OR "primary care" OR "primary healthcare" OR "PHC" OR "primary health") AND (la:("en" OR "pt" OR "es")) AND (year_cluster:[1978 TO 2023]) |

*(Continued)*

**Table 3.** (Continued)

| Database | Search |
|---|---|
| Web of Science | ((ALL = ("political and institutional arrangement" OR "political arrangement" OR "institutional arrangement" OR "political-institutional arrangement" OR "politics" OR "political factor" OR "political system" OR "public policy" OR "governmental organization")) AND ALL = ("financing" OR "resources allocation" OR "spending" OR "expenditure" OR "funding" OR "allocative efficiency" OR "allocation of health resources" OR "national health accounts" OR "health system financing" OR "capitation Fee" OR "pay-for-performance" OR "payment system" OR "fee-for-service plans" OR "capitation" OR "reimbursement" OR "incentive")) AND ALL = ("primary care" OR "primary healthcare" OR "primary health care" OR "primary health" OR "PHC") and English or Portuguese or Spanish (Languages) |
| Scopus | TITLE-ABS-KEY ("political and institutional arrangement" OR "political arrangement" OR "institutional arrangement" OR "political- institutional arrangement" OR "politics" OR "political factor" OR "political system" OR "public policy" OR "governmental organization") AND TITLE-ABS-KEY ("financing" OR "resources allocation" OR "spending" OR "expenditure" OR "funding" OR "allocative efficiency" OR "allocation of health resources" OR "national health accounts" OR "health system financing" OR "capitation Fee" OR "pay-for-performance" OR "payment system" OR "fee-for-service plans" OR "capitation" OR "reimbursement" OR "incentive") AND TITLE-ABS-KEY ("primary care" OR "primary healthcare" OR "primary health care" OR "primary health" OR "PHC") AND PUBYEAR > 1977 AND PUBYEAR < 2024 AND PUBYEAR > 1977 AND PUBYEAR < 2024 AND (LIMIT-TO (LANGUAGE, "English") OR LIMIT-TO (LANGUAGE, "Spanish") OR LIMIT-TO (LANGUAGE, "Portuguese")) |
| Science Direct | "political arrangement" OR "political-institutional arrangement" OR "politics" OR "political factor" OR "political system") Year: 1978–2023 Title, abstract, keywords: ("financing" OR "resources allocation" OR "spending" OR "expenditure" OR "funding" OR "allocation of health resources" OR "health system financing" OR "pay-for-performance" OR "capitation") Title: AND ("primary care" OR "primary healthcare" OR "primary health care" OR "primary health") |

Source: the authors.

## c) Study selection

The database searches are expected to begin in November 2024 and will be performed by a researcher using a software package to organize the search results, which will be recorded and stored in review reports. Duplicate screening and the assessment of the retrieved papers will also be performed using specific software. Title, first, and then abstract screening will be carried out by a team of three reviewers, applying the inclusion and exclusion criteria. In cases of doubt, the team of reviewers will read the full-text version of the publication, applying the same inclusion and exclusion criteria. Any disagreements on study selection will be independently adjudicated by at least one of the other authors.

The review will include studies on primary health care–in particular financing and financial resource allocation–related to the "political dimension" (political factors and arrangements), where the latter is presented as: (1) the problem and/or object of study; (2) explanatory factors derived from methods and approaches in the fields of public health, economics and/or social sciences (notably political science); and (3) discussions that highlight the importance of this dimension in the outcomes of PHC financing models, health policies and the organization of health systems.

In short, the review will include analyses that provide knowledge about governmental and legislative performance and production in health, including factors such as political and ideological orientations, agendas and public policy designs, in addition to intergovernmental relations and between powers, as well as economic regulation and health, lobbies, defence coalitions and representations of interests in the legislative environment. For inclusion of studies, such factors may be evidenced in studies on health system reform processes in which the valorization of PHC as a strategy for expansion, universalization and reformulation is identified.

Concerning financing mechanisms, studies will be included regarding budget composition and execution and spending on PHC, covering the participation of sources of financial resources (public and private), tax revenues and fiscal policy. Regarding allocative decision-making and incentives, the following will be considered: pay-for-performance, capitation, fee-for-service, vouchers, and payment guided by population and equity criteria.

Original research using primary and secondary data and qualitative, quantitative or mixed-method study designs will be included. Essays, opinion and perspective articles, and different types of evidence syntheses (reviews) will also be included.

Publications that do not mention at least one element of the "political dimension" (institutions, actors, disputes, contexts and arenas involved in financial decision-making, resource allocation to PHC, and public health policy, among others) will be excluded. Gray literature, monographs, dissertations, theses, article comments and synopsis/summary, books and book chapters will also not be considered.

The PRISMA flow diagram will be used to present the results of the searches, considering the following stages: identification, screening, eligibility and included studies [38].

Table 4 summarizes the inclusion and exclusion criteria for this scoping review proposal.

**Table 4. Scoping review inclusion and exclusion criteria.**

| Inclusion criteria | Exclusion criteria |
| --- | --- |
| • Studies on primary health care–in particular financing (budget composition and execution and spending, sources of public and private financial resources, tax revenues, fiscal policy) and/or financial resource allocation models (pay-for-performance, capitation, fee-for-service, vouchers, payment guided by population and equity criteria); and–related to the following political factors and arrangements: political and ideological orientations, agendas and designs of government policies, intergovernmental relations and between powers, and/or economic and health regulations, lobbies, defence coalitions and representations of interests in the environment legislative<br>• Studies on policies and health system reform processes in which the valorization of PHC as a strategy for expansion, universalization and reformulation, and which address/discuss its financing (budget composition and execution and spending, sources of public and private financial resources, tax revenues, fiscal policy) and/or financial resource allocation models (pay-for-performance, capitation, fee-for-service, vouchers, payment guided by population and equity criteria); and–related to the following political factors and arrangements: political and ideological orientations, agendas and designs of government policies, intergovernmental relations and between powers, and/or economic and health regulations, lobbies, defence coalitions and representations of interests in the environment legislative<br>• Qualitative studies.<br>• Quantitative studies.<br>• Mixed and/or multi-method studies.<br>• Original peer-reviewed articles presenting results from studies using primary and secondary data.<br>• Essays.<br>• Opinion and perspective articles.<br>• Evidence syntheses (reviews).<br>• Publications in English, Spanish and Portuguese.<br>• Papers published between 1978 and 2023. | • Studies on primary health care that do not touch on financing and/or financial resource allocation.<br>• Studies on primary health care—in particular financing and/or allocation of financial resources models—not related to at least one of political factors and arrangements focused on as inclusion criteria<br>• Books and book chapters.<br>• Book reviews.<br>• Gray literature, government documents and news reports.<br>• Social media content.<br>• Publications not written in English, Spanish or Portuguese.<br>• Papers published before 1978 and after 2023. |

Source: the authors.

## d) Data extraction

As recommended by Pollock *et al* [51], data extraction, analysis, and presentation will be performed by the review team and the process will be recorded in accessible digital documents in line with the research question and objectives of the scoping review. A standardized data extraction form will be created containing the following items: author(s), title, periodical, year, country of origin and type of evidence source.

For each source of evidence, a specific data and information extraction flow will be proposed to previously filter any possible heterogeneity in the structures of the references that will be included. Considering this aspect, and where applicable, the following information will be extracted and, if necessary, organized into subcategories, independently by at least one review author, with general verification by another reviewer: aims, purposes, arguments, problem presented, conceptual framework or theory, study designs, methodological approaches and thematic data, considering the concepts (political factors in financing and allocation of financial resources) and the context (Primary Health Care policies and approaches).

From a thematic point of view, to identify the political factors and arrangements that influence PHC financing and resource allocation, different sections of the publications, preliminarily grouped by source of evidence, will be examined in detail. These factors will be categorized in the light of existing evidence sources to identify possible thematic and methodological gaps in the literature. This review will not assess the quality, use and effectiveness of the evidence.

Throughout the process, doubts, changes in flows and other issues will be discussed by the scope review team. New and different data items, mainly associated with the theme, may be identified, added and categorized, especially when performing qualitative content analysis in an inductive approach. Any such decisions will be recorded and subsequently reported in the presentation of the results of the scoping review.

Table 5 presents the preliminary items of the data extraction form, which will be subjected to a pilot test, for each source of evidence of the articles included to identify needs for adaptation to the review's objectives and questions, possible redundancies, and inaccuracies.

**Table 5. Preliminary items of the data extraction form.**

| Item[a] | Description |
|---|---|
| Author(s) | |
| Title | |
| Periodical | |
| Year of publication | |
| Country/continent | Country<br>Continent |
| Type of evidence source | Original research Review<br>Essay<br>Opinion and perspective article<br>. . . |
| *Aims/purposes/Arguments/Problem*[b] | |
| *Conceptual framework or theory*[b] | |
| *Study design*[b] | Qualitative Quantitative<br>Mixed<br>. . . |
| *Data collection period*[b] */ considered period*[b] | |
| *Methodological approach*[b] | Political analyses<br>Analysis of public health policy<br>. . . |

*(Continued)*

**Table 5.** (Continued)

| Item[a] | Description |
|---|---|
| Thematic data (to guide basic qualitative data analysis) | |
| *PHC financing and financial resource allocation* | PHC financing<br>*Composition and execution of the budget, sources of resources, public and private participation in health spending, fiscal and economic policy*<br>PHC financial resource allocation<br>*Decision-making process, incentives, forms and allocative models: performance, services delivery, capitation, health needs, equity criteria*<br>Primary health care policies and approaches<br>*Focused*<br>*Selective*<br>*Comprehensive*<br>*Public x Private*<br>. . . |
| *Political factors and arrangements* | Political and governmental systems<br>*Political and ideological orientations*<br>*Sectoral and political agendas*<br>*Executive-legislative relations*<br>Legislative environment<br>*Legal and normative, regulatory production, interest representation, lobbies*<br>. . . |

Source: the authors.

[a] Other items may be added to the data extraction form throughout the data extraction process.

[b] When applicable, depending on the source of evidence.

**e) Summary and presentation of results.**   The analysis of the material will involve an initial description using simple frequencies and statistics and thematic categories, through a method of qualitative analysis of thematic content, to identify (frequency and distribution) the main areas, studies, methods, sources of evidence, institutions and academic groups and how they interpret, analyze and approach political factors and arrangements influencing PHC financing and resource allocation, in association with the key concepts and context established in this protocol.

Comprehensive analyzes of specific groups may be performed in an eminently inductive way to distinguish aspects relating specifically to PHC financing models from those relating to financial resource allocation models. The analysis may also identify groups of evidence divided according to the political environment (legislative, executive, and judicial) and preferences and agendas related to the scope of government PHC policies (focused, selective, comprehensive).

Another previously identified possibility is understanding the topic from a historical and institutional perspective, highlighting inflection points during the course of production of knowledge on the topic of interest.

## Supporting information

**S1 Checklist. PRISMA-P 2015 checklist (completed PRISMA-P checklist.docx).**
(DOCX)

## Acknowledgments

We want to thank Adriano da Silva, professional of the Public Health Library of the Oswaldo Cruz Foundation, for his support in designing and improving the search strategies presented in this protocol.

## Author Contributions

**Conceptualization:** Henrique Sant'Anna Dias, Elisabete de Fátima Polo de Almeida Nunes.

**Formal analysis:** Elisabete de Fátima Polo de Almeida Nunes, Caroline Pagani Martins, Marcela Castilho, Fernanda de Freitas Mendonça, Luciana Dias de Lima.

**Methodology:** Henrique Sant'Anna Dias, Adelyne Maria Mendes Pereira, Elisabete de Fátima Polo de Almeida Nunes, Caroline Pagani Martins, Marcela Castilho, Fernanda de Freitas Mendonça, Luciana Dias de Lima.

**Supervision:** Luciana Dias de Lima.

**Validation:** Adelyne Maria Mendes Pereira.

**Writing – original draft:** Henrique Sant'Anna Dias, Adelyne Maria Mendes Pereira, Elisabete de Fátima Polo de Almeida Nunes, Caroline Pagani Martins, Marcela Castilho, Fernanda de Freitas Mendonça, Luciana Dias de Lima.

**Writing – review & editing:** Henrique Sant'Anna Dias, Adelyne Maria Mendes Pereira, Elisabete de Fátima Polo de Almeida Nunes, Caroline Pagani Martins, Marcela Castilho, Fernanda de Freitas Mendonça, Luciana Dias de Lima.

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
