## [Decision Letter · Decision Letter 0]

21 Jun 2024

PONE-D-24-09369Political factors and arrangements influencing primary health care financing and resource allocation: a scoping review protocolPLOS ONE

Dear Dr. Dias,

Thank you for submitting your manuscript to PLOS ONE. After careful consideration, we feel that it has merit but does not fully meet PLOS ONE’s publication criteria as it currently stands. Therefore, we invite you to submit a revised version of the manuscript that addresses the points raised during the review process.

We look forward to receiving your revised manuscript.

Kind regards,

Pengpeng Ye

Academic Editor

PLOS ONE

“This work was supported by the FIOCRUZ Program for Strengthening Primary Health Care in Brazil – PMA 2019 (Fiocruz/VPPCb; grant number 25380.101539/2019- 05)”

Reviewers' comments:

Reviewer's Responses to Questions

**Comments to the Author**

1. Does the manuscript provide a valid rationale for the proposed study, with clearly identified and justified research questions?

Reviewer #1: Partly

Reviewer #2: Yes

2. Is the protocol technically sound and planned in a manner that will lead to a meaningful outcome and allow testing the stated hypotheses?

Reviewer #1: Partly

Reviewer #2: Yes

3. Is the methodology feasible and described in sufficient detail to allow the work to be replicable?

Reviewer #1: Yes

Reviewer #2: Yes

4. Have the authors described where all data underlying the findings will be made available when the study is complete?

Reviewer #1: No

Reviewer #2: Yes

5. Is the manuscript presented in an intelligible fashion and written in standard English?

Reviewer #1: Yes

Reviewer #2: Yes

6. Review Comments to the Author

You may also provide optional suggestions and comments to authors that they might find helpful in planning their study.

Reviewer #1: 1. Although the research questions have been clearly stated, they could be further specified to better guide literature screening and data extraction. For example, it could be made clear which specific political factors and arrangements are the focus of this paper.

2. The inclusion and exclusion criteria could be more detailed. It could be clarified which political factors (e.g. governmental system, fiscal policy, etc.) and resource allocation mechanisms (e.g.capitation, pay-for-performance, etc.).

3.While preliminary items for the data extraction table have been listed, they could be further refined. A breakdown of specific political factors and arrangements could be added, as well as specific indicators and outcomes related to primary health care financing and resource allocation.

4.In the methods section, the process of data extraction and analysis could be described in more detail, especially how to deal with the heterogeneity of multiple study designs and methods, and how to ensure consistency and accuracy of data extraction.

Reviewer #2: The planned protocol manuscript is well presented, some minor issues related to the tentative schedule of data collection and some more clarification about the exact data analysis methods and how the results will be presented need to be mentioned, otherwise this is very intersting to read this protocal and patiently waiting to read the full article when it is done.

7. PLOS authors have the option to publish the peer review history of their article (what does this mean?). If published, this will include your full peer review and any attached files.

Reviewer #1: No

Reviewer #2: No

---

## [Author Response · Author response to Decision Letter 0]

12 Jul 2024

In return for PlosOne's observations raised during the review process, given the submission of the scoping review protocol entitled "Political factors and arrangements influencing primary health care financing and resource allocation: a scoping review protocol, we are happy to present our responses to each point stated in the decision letter. As requested, we uploaded a rebuttal letter named "Response to Reviewers" on the editorial manager platform. 

Kind regards,

Henrique S Dias

---

## [Decision Letter · Decision Letter 1]

30 Jul 2024

Political factors and arrangements influencing primary health care financing and resource allocation: a scoping review protocol

PONE-D-24-09369R1

Dear Dr. Dias,

We’re pleased to inform you that your manuscript has been judged scientifically suitable for publication and will be formally accepted for publication once it meets all outstanding technical requirements.

Kind regards,

Pengpeng Ye

Academic Editor

PLOS ONE

Additional Editor Comments (optional):

Reviewers' comments:

Reviewer's Responses to Questions

**Comments to the Author**

1. Does the manuscript provide a valid rationale for the proposed study, with clearly identified and justified research questions?

Reviewer #1: Yes

2. Is the protocol technically sound and planned in a manner that will lead to a meaningful outcome and allow testing the stated hypotheses?

Reviewer #1: Yes

3. Is the methodology feasible and described in sufficient detail to allow the work to be replicable?

Reviewer #1: Yes

4. Have the authors described where all data underlying the findings will be made available when the study is complete?

Reviewer #1: Yes

5. Is the manuscript presented in an intelligible fashion and written in standard English?

Reviewer #1: Yes

6. Review Comments to the Author

You may also provide optional suggestions and comments to authors that they might find helpful in planning their study.

Reviewer #1: The research topic addressed in this protocol is of significant importance and relevance, particularly in the context of primary healthcare financing and resource allocation. The approach to systematically map political factors and arrangements influencing these aspects is both timely and necessary.

7. PLOS authors have the option to publish the peer review history of their article (what does this mean?). If published, this will include your full peer review and any attached files.

Reviewer #1: No

---

## [Editor Report · Acceptance letter]

2 Aug 2024

PONE-D-24-09369R1 

PLOS ONE

Dear Dr. Dias, 

I'm pleased to inform you that your manuscript has been deemed suitable for publication in PLOS ONE. Congratulations! Your manuscript is now being handed over to our production team.

Kind regards, 

on behalf of

Dr. Pengpeng Ye 

Academic Editor

PLOS ONE